# Determinants of evidence-based practice among health care professionals in Ethiopia: A systematic review and meta-analysis

Amare Zewdie[ID][1]*, Mulat Ayele[2], Tamirat Melis[1], Abebaw Wasie Kasahun[1]

1 Department of Public Health, College of Medicine and Health Science, Wolkite University, Wolkite, Ethiopia, 2 Department of Midwifery, College of Medicine and Health Science, Woldia University, Woldia, Ethiopia

* amarezewdie23@gmail.com

## Abstract

### Introduction

Evidence-based practice (EBP) is the art of using up-to-date information for clinical decision-making. Healthcare professionals at all levels are expected to use the latest research evidence for quality care. In Ethiopia inclusive and nationally representative summarized evidence regarding the level of EBP among health professionals is scarce. Therefore, this systematic review and meta-analysis aimed to assess the pooled prevalence of EBP utilization and its determinants among Ethiopian health professionals.

### Method

A systematic review and meta-analysis were conducted using PRISMA guidelines. Comprehensive literature was searched in PubMed, Google Scholar, and African Online Journal databases. A weighted inverse variance random effect model was used to estimate pooled prevalence. Cochrane Q-test and $I^2$ statistics were computed to assess heterogeneity among studies. Funnel plot and Eggers test were done to assess publication bias. Factors associated with EBP were identified using STATA v. 14.

### Result

Overall, 846 articles were retrieved and finally 23 articles were included in this review. The pooled prevalence of good EBP among health professionals was 52.60% (95%CI; 48.15%-57.05%). Knowledge about EBP (AOR = 2.38, 95% CI: (2.08–2.72)), attitude (AOR = 2.09, 95% CI: (1.67–2.60)), educational status (AOR = 3.12, 95% CI: (2.18–4.47)), work experience (AOR = 2.59, 95% CI: (1.48–4.22)), EBP training (AOR = 2.26, 95% CI: (1.87–2.74)), presence of standard guideline (AOR = 1.94, 95% CI: (1.51–2.50)), internet access (AOR = 1.80, 95% CI: (1.47–2.20)), presence of enough time (AOR = 2.01, 95% CI: (1.56–2.60)) and marital status (AOR = 1.73, 95% CI: (1.32–2.28)) were determinants of EBP.

**Data Availability Statement:** The result of this SRMA was extracted from the data gathered and analyzed based on the stated methods and materials. All the relevant data are within the paper.

**Funding:** The author(s) received no specific funding for this work.

**Competing interests:** The authors have declared that no competing interests exist.

**Abbreviations:** BSc, Bachelor of Science; EBP, Evidence-Based Practice; PRISMA, Preferred Reporting Items for Systematic Reviews and Meta-Analyses; SNNPR, South Nation Nationality and People Region; SRMA, Systematic Reviews and Meta-Analyses.

## Conclusion

Around half of health professionals in Ethiopia have good EBP utilization which was low. Knowledge, attitude, educational status, work experience, EBP training, presence of standard guidelines, internet access, presence of enough time, and single marital status were positively associated with EBP. Therefore future interventions should focus on increasing their knowledge and changing their attitude through providing training and addressing organizational barriers like availing standard guidelines, accessing the internet, and minimizing professionals' workload that enables them to critically appraise and integrate the latest evidence for clinical decision-making to improve the quality of care.

## Introduction

Evidence-based practice (EBP) is an action of reviewing, analyzing, and translating the latest scientific evidence to integrate the best available research evidence to clinical experience and patient preference in the actual clinical setup [1]. The fundamental principle of EBP is the use of up-to-date information for clinical decision-making [2]. This approach of using the latest research evidence for clinical practice can not only improve the care and treatment of patients rather it also leads to the discovery of effective drugs and advanced medical technologies which can increase the quality of healthcare delivery [3]. EBP in health system management is also the base for policy formulation and evaluation. Health researchers have a role to support policy-makers with appropriate evidence that can improve the quality of care being delivered [4].

The significance of EBP is multidirectional; thus from the professional side EBP also positively influences the practice of health care professionals and enables them to shift from intuition and tradition to scientifically valid and reliable practices [5]. EBP also results in high job satisfaction, increases work efficiency, and fills the gap between research, theory, and practice [6]. Moreover, evidence-based medical care can improve professionalism and continuous professional development in the healthcare workforce [7].

As improved outcomes are being observed in patients receiving evidence-based medical care; currently, different international organizations as well as countries' government bodies recognize the significance of EBP and strive to achieve the use of the latest health research evidence for clinical practice and also for health policy formulation and program implementation [8]. Even though evidence-based medical practice is significantly important for low-income countries; since it is the most cost-effective and efficient use of scarce healthcare resources however its implementation faced several challenges [9, 10].

Researchers have identified several barriers to EBM implementation in low-resource settings [11]. In Ethiopia Despite the effort to enhance EBP implementation is underway, its execution in the actual setup has confronted several barriers. Healthcare professionals also continue to provide care as previously without the integration of EBP [12]. A systematic review in the country also identified; a lack of EBP knowledge, inadequate resources and time, lack of EBP training and management support, difficulty in interpreting research findings, overloading of patients, and negative attitudes as the most frequent barriers to EBP implementation [13].

So far, in Ethiopia, different studies have been done on health professionals' EBP utilization and found a highly variable level of utilization across the regions of the country. Although there was one systematic review and meta-analysis on EBP utilization of Ethiopian health professionals [14] it was not inclusive or is shallow and does not show any sort of subgroup

analysis since it includes only eight articles despite there being more than twenty primary studies on the topic. Thus the previous review lacks representativeness despite the need to summarize the issue as a nation and is expected to provide inclusive, representative, and latest evidence for decision-making. Therefore this systematic review and meta-analysis aimed to assess the pooled prevalence of EBP utilization and identify its associated factors among Ethiopian health professionals.

## Method

### Study design and setting

A systematic review and meta-analysis were conducted on determinants of EBP among health professionals in Ethiopia. Preferred Reporting Items for Systematic Review and Meta-Analysis guidelines were followed (S1 Table). PRISMA is a protocol consisting of checklists that guide the conduct and reporting of systematic reviews and meta-analyses, which increase the transparency and accuracy of reviews in medicine and other fields [15]. Ethiopia is one of the low-income countries located in the Horn of Africa with a 2022 projected population of 123.4 million, 133.5 million in 2032, and 171.8 million in 2050 [16]. For administrative purposes, Ethiopia is divided into 11 regions and 2 city administrations. Regions are further classified into zones, and zones are divided into districts. Finally, districts are divided into kebele (the smallest administrative division contains 2000 up to 3500 residents).

### Search strategies and sources of information

We have checked the PROSPERO database (http://www.library.ucsf.edu/) whether published or ongoing projects exist related to the topic to avoid any further duplication. Thus, the findings revealed that there were no ongoing or published registered reviews in the area of this topic. Then this systematic review and meta-analysis were registered in the PROSPERO database with Id no of CRD42023407822. Comprehensive literature was searched using international databases PubMed, Google Scholar, and African Online Journal to retrieve related articles from March 01 to 10, 2023. Grey literature was searched using Google. Search terms were formulated using PICO guidelines through online databases. Medical Subject Headings (MeSH) and key terms had been developed using different Boolean operators 'AND' and 'OR'. The following search term was used: "Evidence-based practice" OR "Evidence-based medicine" OR "Evidence-based treatment approach" AND "Health professionals" OR "Health workers" OR "medical practitioners" AND Ethiopia.

### Eligibility criteria

To be included in this systematic review and meta-analysis, studies should be on evidence-based practice of Ethiopian health professionals and its determinants in the English language, without restriction on race, sex, or publication date (until the last search date March 10, 2023). Articles without full abstracts or texts and articles reported out of the outcome interest were excluded. Citations without abstracts and/or full-text, commentaries, anonymous reports, letters, editorials, reviews, and meta-analyses were excluded at each respective stage of screening.

### Outcome measurements

This study has two main outcomes. The primary outcome was the magnitude of evidence-based practice of health professionals. It is defined as the proportion of participants who follow good evidence-based clinical practice. Therefore, all included primary studies measure the study participants' level of evidence-based practice and are categorized as having good

evidence practice and poor. Then, the response was analyzed and presented as the magnitude of evidence-based practice of health professionals. The secondary outcomes were factors associated with the evidence based practice of health professionals.

## Data extraction

All studies obtained from the considered databases were exported to Endnote version X8 software to remove duplicate studies. Then after, all studies were exported to a Microsoft Excel spreadsheet. All authors independently extracted the important data using a standardized data extraction form which was adapted from the Joanna Briggs Institute (JBI) data extraction format. For the first outcome (prevalence) the data extraction format included (primary author, year of publication, regions, study area, sample size, and prevalence of EBP with 95% CI). We extracted data for the second outcome (associated factors to EBP) using a 2 by 2 table format. Finally, the log odds ratio for each factor was calculated using STATA v 14.

## Quality assessment

To assess the quality of each study included in this systematic review and meta-analysis, the modified Newcastle Ottawa Quality Assessment Scale for cross-sectional studies was used [17] (S2 Table). Two Authors (AZ, AWK) assessed the quality of each study (i.e. methodological quality, sample selection, sample size, comparability and the outcome, and statistical analysis of the study). In the case of disagreement between two authors; another two authors (MA, TM) were involved and discussed and resolved the disagreement.

## Data processing and analysis

The extracted Microsoft Excel spreadsheet format data was imported to STATA version 14 for analysis. Then weighted inverse variance random effect model was used to estimate the pooled prevalence of the evidence-based practice of health professionals in Ethiopia. Cochrane Q-test and $I^2$ statistics were computed to assess heterogeneity among all studies. Accordingly, if the result of $I^2$ is 0% to 40% it is mild heterogeneity, 40 to 70% would be moderate heterogeneity, and 70 to 100% would be considerable heterogeneity [18]. Funnel plot and Eggers test were done to assess publication bias. The p-value >0.05 indicated that there was no publication bias. Subgroup analysis was done based on the study region. A forest plot format was used to present the pooled prevalence of the evidence-based practice of health professionals with 95% CI. Factors associated with evidence-based practices' of health professionals were also identified using STATA v 14.

## Result

Overall, 846 articles were retrieved using our search strategy. International databases; PubMed, Google Scholar, and African Journals Online were searched. Duplicates (394) were removed and 452 articles remained. After reviewing, (n = 348) articles were excluded by title, and (n = 56) articles were excluded by reading abstracts. Therefore, 48 full-text articles were accessed and assessed for inclusion criteria, resulting in the further exclusion of 25 articles due to the mentioned reason. As a result, 23 studies fulfilled the inclusion criteria to undergo the final systematic review and meta-analysis (Fig 1).

Of the included studies in this SRMA, eleven were done in Amhara, six in the Oromia region, four in Addis Ababa, and two in SNNPR. All included studies were cross-sectional. Regarding quality of included studies, the Newcastle Ottawa Quality Assessment scale score of all included studies lies from 8 to 9 which is good (Table 1).

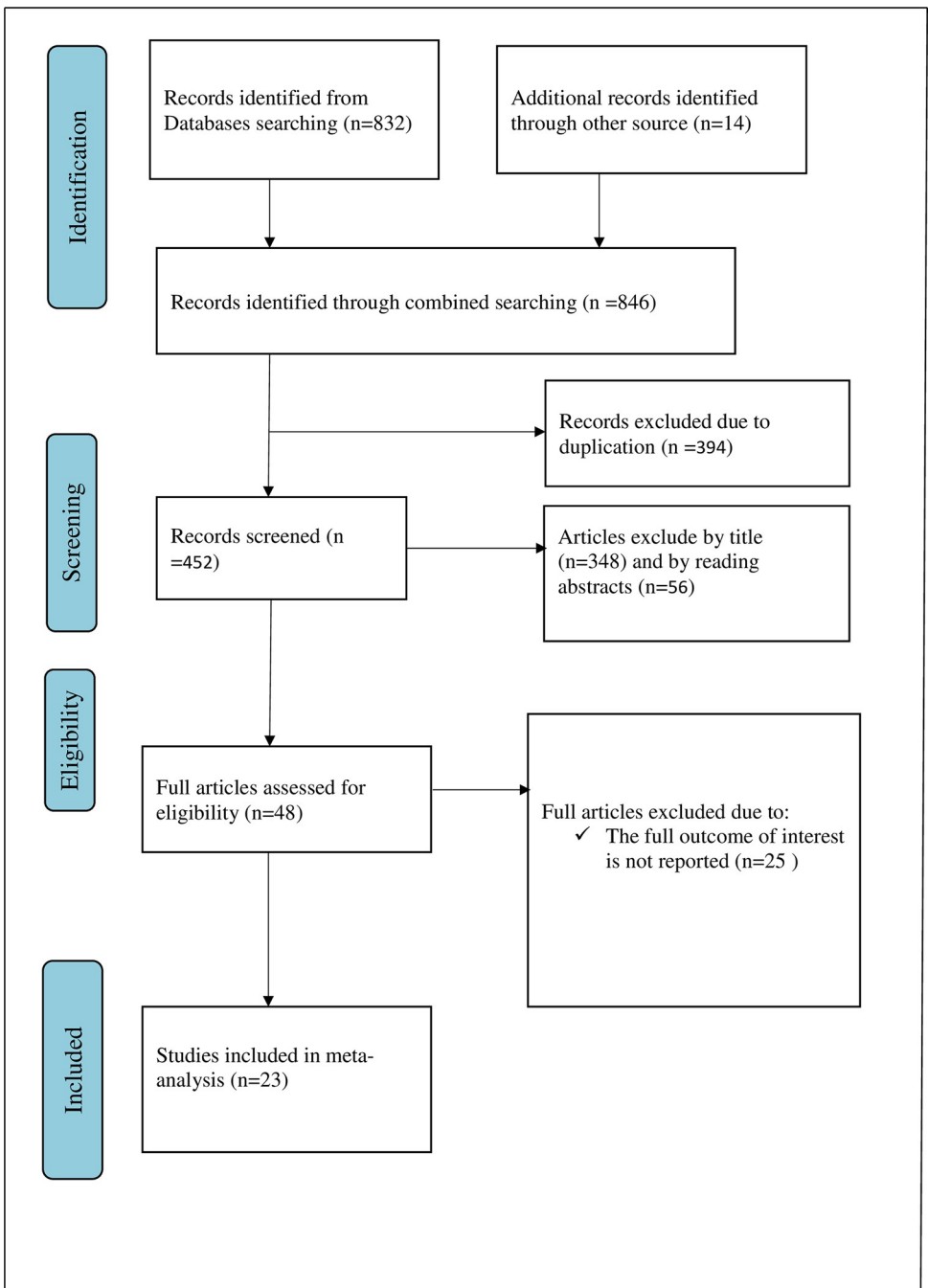

**Fig 1. Flow chart of study selection for systematic review and meta-analysis on EBP utilization of health professionals and its determinant in Ethiopia, 2023.**

## The magnitude of EBP utilization among healthcare professionals in Ethiopia

The pooled prevalence of good EBP utilization of health professionals in Ethiopia was 52.60% (95%CI; 48.15%-57.05%), with the Cochrane heterogeneity index ($I^2$ = 94.3%), P = 0.000, showing substantial heterogeneity of different studies ($I^2$>70%). The finding was presented using a forest plot (Fig 2).

**Table 1. Characteristics of included studies in the systematic review and meta-analysis on EBP of health professionals and its determinants in Ethiopia.**

| S.no | Author | Year | Region | Study design | Sample size | Magnitude of EBP | Study quality |
|------|--------|------|--------|--------------|-------------|------------------|---------------|
| 1 | Alemayehu, et al. [19] | 2021 | SNNPR | Crossect | 671 | 55.0% | Good |
| 2 | Tadesse, et al. [20] | 2018 | SNNPR | Crossect | 208 | 61.54% | Good |
| 3 | Abera, et al. [21] | 2015 | Oromia | Crossect | 115 | 65.22% | Good |
| 4 | Dereje, et al. [22] | 2018 | Oromia | Crossect | 253 | 51.78% | Good |
| 5 | Hoyiso, et al. [23] | 2015 | Oromia | Crossect | 302 | 66.22% | Good |
| 6 | Megersa, et al. [24] | 2021 | Oromia | Crossect | 403 | 52.36% | Good |
| 7 | Wodajo, et al. [25] | 2022 | Oromia | Crossect | 278 | 66.67% | Good |
| 8 | Worku, et al. [26] | 2017 | Oromia | Crossect | 124 | 32.26% | Good |
| 9 | Alene, et al. [27] | 2020 | Addis Ababa | Crossect | 135 | 85.19% | Good |
| 10 | Assefa, et al. [28] | 2020 | Addis Ababa | Crossect | 422 | 56.87% | Good |
| 11 | Hadgu, et al. [29] | 2015 | Addis Ababa | Crossect | 210 | 57.62% | Good |
| 12 | Mitiku, et al [30] | 2020 | Addis Ababa | Crossect | 386 | 51.04% | Good |
| 13 | Aynalem, et al. [31] | 2019 | Amhara | Crossect | 671 | 55.0% | Good |
| 14 | Beshir, et al. [32] | 2015 | Amhara | Crossect | 431 | 52.90% | Good |
| 15 | Dagne, et al. [33] | 2019 | Amhara | Crossect | 790 | 34.81% | Good |
| 16 | Debeb, et al. [34] | 2021 | Amhara | Crossect | 391 | 54.73% | Good |
| 17 | Degu, et al. [35] | 2020 | Amhara | Crossect | 507 | 47.14% | Good |
| 18 | Dessie, et al. [36] | 2017 | Amhara | Crossect | 405 | 40% | Good |
| 19 | Kassahun, et al. [37] | 2015 | Amhara | Crossect | 207 | 38.16% | Good |
| 20 | Melesew, et al. [38] | 2021 | Amhara | Crossect | 385 | 55.32% | Good |
| 21 | Wassie, et al. [39] | 2014 | Amhara | Crossect | 169 | 40.83% | Good |
| 22 | Yehualashet,et al. [40] | 2020 | Amhara | Crossect | 403 | 48.39% | Good |
| 23 | Yideg, et al. [41] | 2022 | Amhara | Crossect | 406 | 44.09% | Good |

## Publication bias

In this systematic review and meta-analysis, a funnel plot was done to check the presence of publication bias at a significance level of less than 0.05. The Egger's regression test was not statistically significant P = 0.266 (p>0.05) confirming no evidence of publication bias, as presented by the funnel plot (Fig 3).

## Subgroup analysis of EBP utilization among health care professional in Ethiopia

The finding of subgroup analysis by region showed that the pooled prevalence of good EBP utilization of health professionals was highest in Addis Ababa (62.63%; 95% CI: (48.24–77.02), $I^2$ = 96.3%, p = 0.000), followed by SNNPR (57.68%; 95% CI: (51.37–63.99), $I^2$ = 64.8%, p = 0.092), then Oromia (55.40%; 95% CI: (46.55–64.25), $I^2$ = 91.8%, p = 0.000) and least Amhara (46.57%; 95% CI: (41.77–51.37), $I^2$ = 91.2%, p = 0.000) (Fig 4).

## Sensitivity analysis

A random-effect model result showed that; no single study has influenced the overall pooled prevalence of EBP of health professionals in Ethiopia since all estimate lies inside the confidence interval.

## Determinants of EBP Utilization of health professionals in Ethiopia

In this systematic review and meta-analysis variables which are associated in two or more primary studies which report adjusted odd ratio which is output of multiple logistic regression

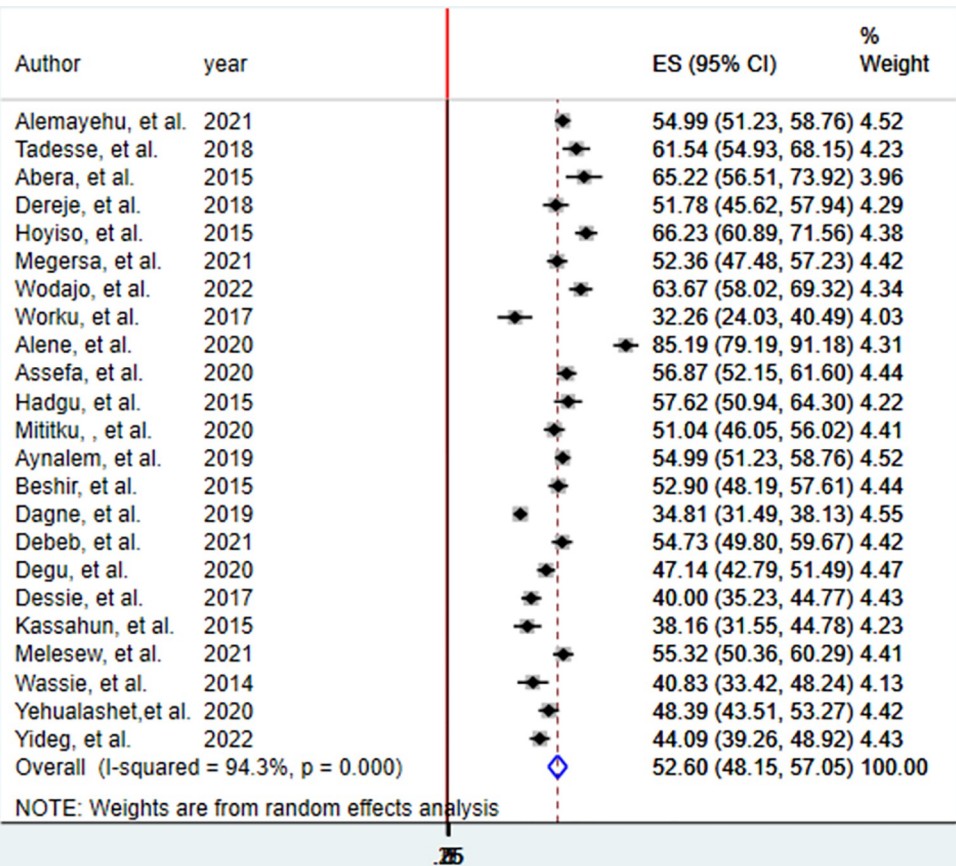

**Fig 2. The pooled prevalence of good EBP utilization of health professionals in Ethiopia, 2023.**

are considered in identification of the overall determinants of EBP utilization. Accordingly, knowledge about EBP, attitude, educational status, work experience, EBP training, presence of standard guidelines, internet access, presence of enough time, and marital status were significantly associated with health professional EBP utilization in Ethiopia. Healthcare professionals who have good knowledge about EBP were 2.38 times more likely to have good EBP utilization as compared to their counterparts (AOR = 2.38, 95% CI: (2.08–2.72)). Similarly, Health professionals who have a positive attitude to EBP were twice more likely to have good EBP as compared to professionals who have a negative attitude (AOR = 2.09, 95% CI: (1.67–2.60)). Professionals having BSc and above educational qualifications were more than 3 times more likely to have good EBP as compared to diploma professionals (AOR = 3.12, 95% CI: (2.18–4.47)). Regarding work experience, health professionals who have long work experience were 2.59 times more likely to have good EBP as compared to their counterparts (AOR = 2.59, 95% CI: (1.48–4.22)). Moreover, health professionals who have EBP training, standard guidelines at the workplace, internet access, enough time, and single in marital status were 2.26, 1.94, 1.80, 2.01, and 1.73 times more likely to have good evidence-based practice as compared to their counterparts respectively (Table 2).

## Discussion

Evidence-based practice is nowadays one of the approaches which are continuously promoted in the current health care system as an essential component to improve the quality of health

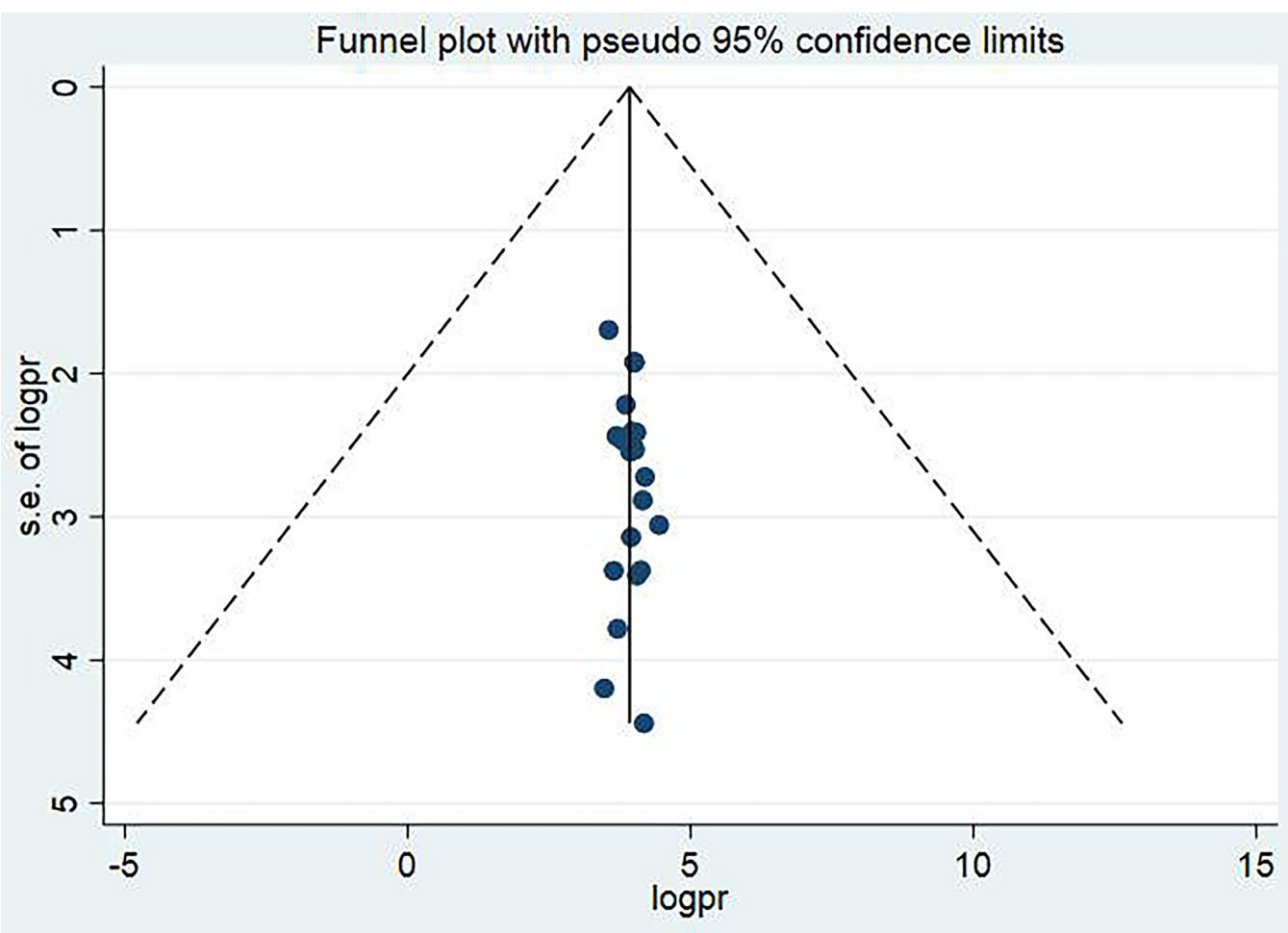

**Fig 3. Funnel plot showing the symmetric distribution of articles on EBP utilization of health professionals and its determinant in Ethiopia, 2023.**

care being delivered. Healthcare professionals at all levels are expected to utilize the latest research evidence in clinical decision-making for better patient care, especially in low-income countries for better use of scarce resources in the setup. Ethiopia is one of the developing countries in which its administrative bodies recognized the best use of up-to-date clinical evidence for patient care. The current systematic review and meta-analysis aimed to assess the magnitude and determinants of EBP of health professionals in Ethiopia. Accordingly, the pooled prevalence of good EBP utilization among health professionals in Ethiopia was 52.60% (95% CI; 48.15%-57.05%). The finding was consistent with studies done in Kenya (53.6%) [42], Zambia (54%) [43], and Jordan (56.1%) [44]. The finding is lower than study's findings of Nigeria (78.7%) [45] and Uganda [46]; the possible discrepancy may be the difference in EBP awareness and integration of EBP in the teaching curriculum of health professionals. The result implies effort should be made in increasing health professionals' habit of using the latest research evidence for clinical practice so as to deliver quality health service.

The current review also identified the factor significantly associated with health professionals' EBP utilization. Thus, health professionals who have good knowledge about EBP were more than twice more likely to have good EBP as compared to health professionals who have poor knowledge. The finding was in line with studies conducted in Kerman University of Iran

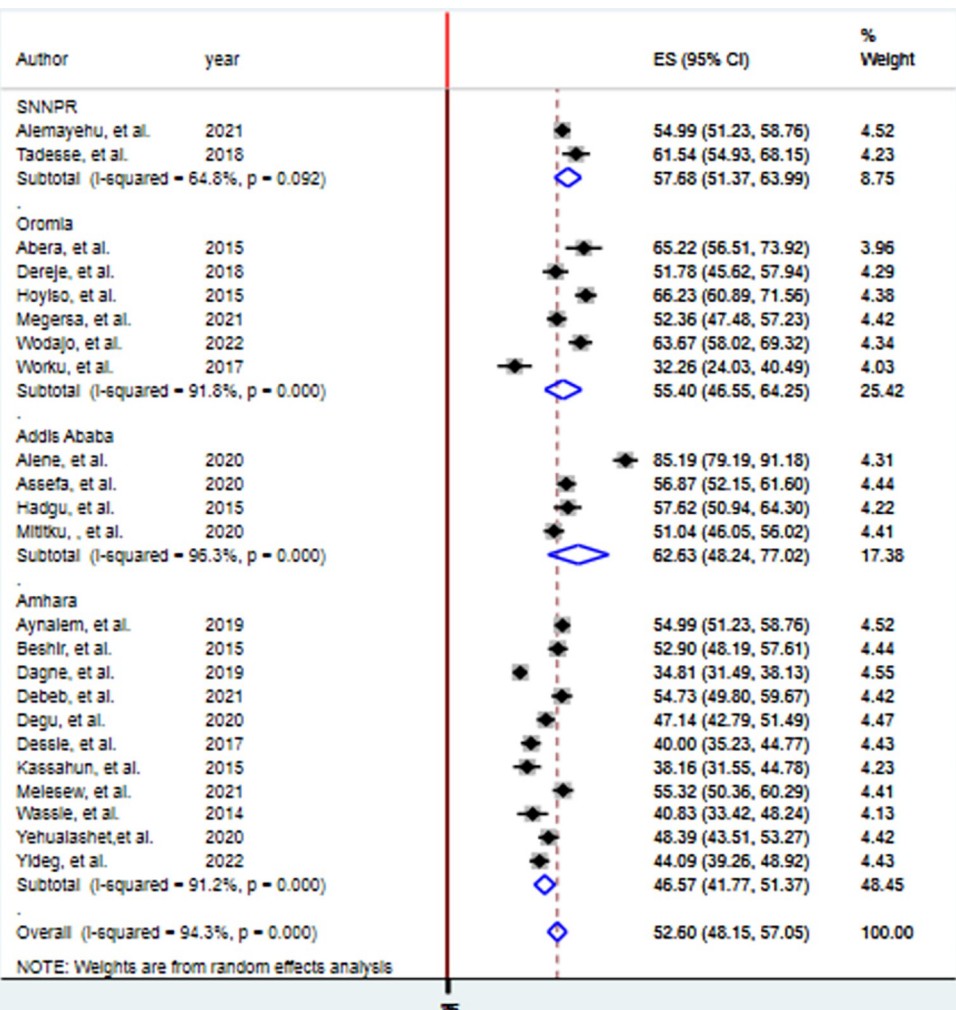

**Fig 4. Forest plot showing subgroup analysis of EBP utilization of health professionals in Ethiopia, 2023.**

[47], Saudi Arabia [48], and Norway [49] in which good knowledge about EBP was positively linked with better EBP utilization. This might be due to the fact that having good knowledge about EBP not only improves the confidence of health professionals, but also enhances skills to implement it. Similarly health professionals who took EBP training have more than two times good EBP as compared to health professionals who haven't taken the training. The evidence implies improving EBP-related awareness of health professionals through focused training can significantly increase health professionals' EBP utilization.

In our review, health professionals who have positive attitudes were around twice more likely to implement EBP as compared to their counterparts. The result was consistent with studies in Saudi Arabia [48] and Norway [49] in which health professionals' favorable attitude towards EBP and its positive outcomes can further motivate them for better EBP utilization. This is due to the positive feeling to outcomes of EBP improve the readiness of health professionals to review, appraise, integrate, and incorporate the available evidence for clinical decisions. Therefore, future interventions should influence professionals' feelings about the advantages of EBP in improving clinical care and patient outcomes later to improve their practice.

**Table 2. Factors associated with EPB utilization of health professionals in Ethiopia.**

| Variable | Authors | AOR | 95%CI | Pooled AOR | 95%CI of pooled AOR |
|---|---|---|---|---|---|
| Good knowledge | Alemayehu, et al. | 2.044 | 1.406–2.972 | 2.38 | 2.08–2.72 |
| | Beshir, et al. | 1.612 | 1.06–2.45 | | |
| | Aynalem, et al. | 2.044 | 1.406–2.972 | | |
| | Dagne, et al. | 3.06 | 1.6–5.77 | | |
| | Debeb, et al. | 2.1 | 1.3–3.38 | | |
| | Dereje, et al. | 2.084 | 1.118–3.886 | | |
| | Hadgu, et al. | 3.2 | 1.5–7 | | |
| | Kassahun, et al. | 5.3 | 2–13.9 | | |
| | Megersa, et al. | 1.785 | 1.13–2.82 | | |
| | Wodajo, et al. | 2.95 | 1.52–5.73 | | |
| | Mititku, et al | 2.81 | 1.79–4.37 | | |
| | Wassie, et al. | 2.22 | 1.1–4.49 | | |
| | Yehualashet,et al | 1.86 | 1.22–2.835 | | |
| | Yideg, et al | 7.95 | 4.83–13.07 | | |
| Positive attitude | Dagne, et al. | 5.02 | 1.2–21.5 | 2.09 | 1.67–2.60 |
| | Degu, et al. | 1.8 | 1.24–2.62 | | |
| | Kassahun, et al. | 3.34 | 1.3–8.6 | | |
| | Wodajo, et al. | 3.13 | 1.59–6.16 | | |
| | Mititku, et al | 1.8 | 1.14–2.86 | | |
| | Yehualashet, et al. | 2.05 | 1.318–3.193 | | |
| BSc degree and above | Degu, et al. | 2.15 | 1.15–4.02 | 3.12 | 2.18–4.47 |
| | Dereje, et al. | 3.186 | 1.634–6.21 | | |
| | Wodajo, et al. | 5.75 | 2.23–14.84 | | |
| | Mititku, et al | 4.09 | 1.45–11.55 | | |
| | Yideg, et al. | 3.05 | 1.08–8.67 | | |
| Work experience | Debeb, et al. | 2.13 | 1.21–3.73 | 2.59 | 1.48–4.22 |
| | Tadesse, et al. | 13.799 | 2.352–80.974 | | |
| EBP training | Alemayehu, et al. | 3.224 | 1.957–5.311 | 2.26 | 1.87–2.74 |
| | Beshir, et al. | 1.906 | 1.223–2.97 | | |
| | Aynalem, et al | 3.224 | 1.957–5.311 | | |
| | Debeb, et al | 1.81 | 1.12–2.93 | | |
| | Kassahun, et al. | 4.5 | 1.61–12.71 | | |
| | Mititku, et al | 1.73 | 1.12–2.67 | | |
| | Yideg, et al | 2.13 | 1.26–3.58 | | |
| Presence of standard guideline | Alemayehu, et al. | 1.827 | 1.249–2.673 | 1.94 | 1.51–2.50 |
| | Wodajo, et al. | 2.88 | 1.46–5.7 | | |
| | Aynalem, et al. | 1.827 | 1.249–2.673 | | |
| Internet access | Alemayehu, et al. | 1.655 | 1.119–2.448 | 1.80 | 1.47–2.20 |
| | Beshir, et al. | 1.831 | 1.191–2.816 | | |
| | Aynalem, et al | 1.655 | 1.119–2.448 | | |
| | Wassie, et al. | 2.43 | 1.12–5.29 | | |
| | Yideg, et al. | 2.02 | 1.25–3.27 | | |
| Presence of enough time | Beshir, et al. | 1.698 | 1.122–2.57 | 2.01 | 1.56–2.60 |
| | Hadgu, et al. | 7.9 | 3.5–17.6 | | |
| | Yehualashet, et al | 1.67 | 1.065–2.627 | | |
| | Yideg, et al. | 1.9 | 1.09–3.31 | | |

(*Continued*)

**Table 2.** (Continued)

| Variable | Authors | AOR | 95%CI | Pooled AOR | 95%CI of pooled AOR |
|---|---|---|---|---|---|
| Single marital status | Alemayehu, et al | 1.662 | 1.089–2.536 | 1.73 | 1.32–2.28 |
| | Wassie, et al. | 2.21 | 1.08–4.51 | | |
| | Aynalem, et al | 1.662 | 1.089–2.536 | | |

Moreover, individual factors such as educational qualification level and work experience also show statistically significant associations with professionals' EBP. Hence health professionals who have BSc and above educational qualifications and also professionals who have five years and above work experience have better EBP utilization as compared to professionals who have a diploma and have less than five-year work experience respectively. The finding was in line with the study finding of Israel [50] in which professionals who have higher educational status were more likely to undergo EBP implementation. This may be due to professionals having higher education levels and having greater years of experience being more technologically oriented, thus improving search strategies, or they are more exposed to the integration of EBP in their courses and teaching programs. The finding implies intervention in improving the EBP of health professionals should target diploma health professionals and fresh graduates and professionals with few years of experience through onsite training or experience sharing from senior staff.

Furthermore in this review health professionals who are single in marital status were seventy-three percent more likely to have good EBP implementation as compared to married professionals. This might be due to the fact that married persons have additional responsibilities and workloads which make them very busy and difficult to implement EBP in day-to-day clinical practice.

Organizational or work environment-related factors were also significantly associated with the EBP of Ethiopian healthcare professionals. Since health professionals whose work environment has standard guidelines, internet access, and enough time for evidence searching were more likely to implement EBP as compared to their counterparts. The finding was supported by evidence from different parts of the world [51, 52] in which a work environment that has standard guidelines for medical practice, the internet for evidence searching, and professionals who are not overloaded or have enough time for evidence appraisal were suitable for EBP implementation. The finding suggests future interventions for improving evidence-based medicine should focus on addressing those organizational barriers through availing standard guidelines at the workplace, accessing the internet in the work environment, and minimizing professional workload that helps them to critically appraise and integrate the latest research evidence for clinical decision making.

Not standing with its finding, this SRMA has limitation. Our search strategy found primary studies which include study participants from different fields such as nurses, midwifes, medical doctors and other health professionals which makes difficult to infer the finding for specific health professional rather for all health professionals as a whole.

## Conclusion

Around half of health professionals in Ethiopia have good EBP utilization which was low, which indicates the need to raise the habit of using the latest research evidence in clinical decision-making for better client care. Good knowledge about EBP, positive attitude, higher educational status, greater work experience, EBP training, presence of standard guidelines, internet access, presence of enough time, and single marital status was positively associated

with good EBP utilization. Therefore intervention efforts should focus on increasing EBP awareness and changing attitudes, especially targeting health professionals who are lower in educational status, less experienced, and single in marital status through providing training and standard guidelines, accessing the internet, and enough time for evidence searching to increase their evidence-based clinical practice so as to improve the quality of care.

## Supporting information

**S1 Table. PRISMA 2020 checklist followed for this systematic review and meta-analysis.**
(DOCX)

**S2 Table. Newcastle-Ottawa Quality Assessment Scale for cross sectional studies used in the systematic review and meta-analysis 2022.**
(DOCX)

## Acknowledgments

We would like to thank all authors of the primary studies which are included in this systematic review and meta-analysis.

## Author Contributions

**Conceptualization:** Amare Zewdie.

**Data curation:** Amare Zewdie, Mulat Ayele, Tamirat Melis, Abebaw Wasie Kasahun.

**Formal analysis:** Amare Zewdie, Tamirat Melis, Abebaw Wasie Kasahun.

**Funding acquisition:** Amare Zewdie.

**Investigation:** Amare Zewdie, Mulat Ayele, Abebaw Wasie Kasahun.

**Methodology:** Amare Zewdie, Mulat Ayele, Tamirat Melis, Abebaw Wasie Kasahun.

**Project administration:** Amare Zewdie, Mulat Ayele, Abebaw Wasie Kasahun.

**Resources:** Amare Zewdie, Abebaw Wasie Kasahun.

**Software:** Amare Zewdie, Abebaw Wasie Kasahun.

**Supervision:** Amare Zewdie, Abebaw Wasie Kasahun.

**Validation:** Amare Zewdie, Tamirat Melis, Abebaw Wasie Kasahun.

**Visualization:** Amare Zewdie, Mulat Ayele, Tamirat Melis, Abebaw Wasie Kasahun.

**Writing – original draft:** Amare Zewdie, Mulat Ayele.

**Writing – review & editing:** Amare Zewdie, Mulat Ayele, Tamirat Melis, Abebaw Wasie Kasahun.

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
