## [Decision Letter · Decision Letter 0]

23 May 2023

PONE-D-23-09153Determinants of evidence-based practice utilization among health care professionals in Ethiopia: a systematic review and meta-analysisPLOS ONE

Dear Dr. Zewdie,

Thank you for submitting your manuscript to PLOS ONE. After careful consideration, we feel that it has merit but does not fully meet PLOS ONE’s publication criteria as it currently stands. Therefore, we invite you to submit a revised version of the manuscript that addresses the points raised during the review process.

We look forward to receiving your revised manuscript.

Kind regards,

Mulu Tiruneh

Academic Editor

PLOS ONE

file:///home/nkw-ld22-073/Downloads/journal.pone.0269273.pdf

https://www.researchgate.net/publication/49793517_Evidence-based_medical_practice_in_developing_countries_The_case_study_of_Iran

In your revision ensure you cite all your sources (including your own works), and quote or rephrase any duplicated text outside the methods section. Further consideration is dependent on these concerns being addressed.

Reviewers' comments:

Reviewer's Responses to Questions

**Comments to the Author**

1. Is the manuscript technically sound, and do the data support the conclusions?

Reviewer #1: Yes

Reviewer #2: Yes

2. Has the statistical analysis been performed appropriately and rigorously? 

Reviewer #1: Yes

Reviewer #2: Yes

3. Have the authors made all data underlying the findings in their manuscript fully available?

Reviewer #1: No

Reviewer #2: Yes

4. Is the manuscript presented in an intelligible fashion and written in standard English?

Reviewer #1: Yes

Reviewer #2: Yes

5. Review Comments to the Author

Reviewer #1: The manuscript has greatly improved and fit for publication. Howerver, under the results section, the quotes did not reflect qoutes.

Secondly, the 2nd theme (Shift in Prosocial Behaviour Over the Pandemic) was too long.

In overral, thank you

Reviewer #2: This submission presents the results of determinants of evidence-based practice utilization among health care professionals in Ethiopia: a systematic review and meta-analysis. The research question was interesting and has public health relevance even though I have some queries regarding the manuscript.

Specific comments/questions are listed below.

1. In affilation number 2 the word “Woldia” was not correctly spelled.

2. The whole text needs to be written in correct English, to prevent misunderstanding.

3. I think the title should be re-written again since the use of words like “ practice” and “utilization” consequatively is ambigious.

4. The last statement of your conclusion in the abstract should be re-written.

5. In your discussion of the pooled prevalence you have reason out the availability teaching curriculum of health professionals in Nigeria and Uganda increases the EBP; what is the authors' reference for this?

6. “The finding suggest future intervention for improving evidence based medicine should focus on addressing those organizational barriers through availing standard guideline at work place, accessing internet in the work environment and minimizing professional workload that helps them to critically appraise and integrate latest research evidence for clinical decision making.” This statement at the end of the discussion should not be the part of the abstract.

6. PLOS authors have the option to publish the peer review history of their article (what does this mean?). If published, this will include your full peer review and any attached files.

Reviewer #1: No

Reviewer #2: **Yes: **Fentaw Teshome Dagnaw

---

## [Author Response · Author response to Decision Letter 0]

24 May 2023

A point-by-point response to reviewers and editor

First of all, we would like to say thank you to both the reviewers and editor, we are so lucky to have you who had a great contribution to the improvement of this work. Saying this we have addressed the given comments as follows.

Reviewer #2

Comment: In affilation number 2 the word “Woldia” was not correctly spelled.

Response: ok thank you for your comment. We would like to say sorry for the spelling error we have made in the institution naming. We accept the reviewer's comment and we will correct it in the revised manuscript as shown in the track change. 

Comment: The whole text needs to be written in correct English, to prevent misunderstanding.

Response: ok thank you for your comment. We accept the reviewer's comment and try to edit the revised document by consulting an English-proficient individual to avoid misunderstandings that can arise from grammar, punctuation, and spelling errors.

Comment: I think the title should be re-written again since the use of words like “ practice” and “utilization” consequatively is ambigious.

Response: ok thank you for your comment. We accept your comment and rewrite it as shown in the track change of the revised manuscript.

Comment: The last statement of your conclusion in the abstract should be re-written.

Response: ok thank you for your comment. We accept your comment and rewrite it as shown in the track change of the revised manuscript.

Comment: In your discussion of the pooled prevalence you have reason out the availability teaching curriculum of health professionals in Nigeria and Uganda increases the EBP; what is the authors' reference for this?

Response: ok thank you for the question raised. Our justification for lower EBP in Ethiopia as compared to health professionals in Nigeria and Uganda is the difference in EBP awareness and integration of EBP in the teaching curriculum of health professionals. Our reference for this is the cited article in the comparison. While we read the cited article it says most (90.2) of the respondents have good knowledge about evidence-based practice that might be acquired through curriculum-based training in their formal education.

Comment: “The finding suggest future intervention for improving evidence based medicine should focus on addressing those organizational barriers through availing standard guideline at work place, accessing internet in the work environment and minimizing professional workload that helps them to critically appraise and integrate latest research evidence for clinical decision making.” This statement at the end of the discussion should not be the part of the abstract.

Response: ok thank you for your comment we accept the reviewer's suggestion and we will put it in the abstract in the conclusion subsection as you suggested above in the revised manuscript as shown in the track change.

---

## [Editor Report · Decision Letter 1]

29 May 2023

PONE-D-23-09153R1Determinants of evidence-based practice among health care professionals in Ethiopia: a systematic review and meta-analysisPLOS ONE

Dear Dr. Zewdie,

Thank you for submitting your manuscript to PLOS ONE. After careful consideration, we feel that it has merit but does not fully meet PLOS ONE’s publication criteria as it currently stands. Therefore, we invite you to submit a revised version of the manuscript that addresses the points raised during the review process.

We look forward to receiving your revised manuscript.

Kind regards,

Mulu Tiruneh

Academic Editor

PLOS ONE
---

## [Author Response · Author response to Decision Letter 1]

31 May 2023

A point-by-point response to reviewers and editor

First of all, we would like to say thank you to both the reviewers and editor, we are so lucky to have you who had a great contribution to the improvement of this work. Saying this we have addressed the given comments as follows.

Journal requirements

please review your reference list to ensure that it is complete and correct. If you have cited papers that have been retracted, please include the rationale for doing so in the manuscript text, or remove these references and replace them with relevant current references. Any changes to the reference list should be mentioned in the rebuttal letter that accompanies your revised manuscript. If you need to cite a retracted article, indicate the article’s retracted status in the References list and also include a citation and full reference for the retraction notice.

Response: Ok thank you for the suggestion. We have reviewed our reference and corrected as suggested above. As our study is systematic review and meta-analysis to minimize publication bias we have included primary studies which are not published. So it seems retracted articles but not. To solve this we have indicated the URL of the web page in which the studies were available. We have put the URL for reference 27, 28, 30, 41, 43 as they were not published but available at research repositories and other webpages and indicate reference 38 as it was a thesis. Additionally reference 20 and 45 were not correctly cited in the previous manuscript version and we have correctly cited those references.

---

## [Decision Letter · Decision Letter 2]

21 Aug 2023

PONE-D-23-09153R2Determinants of evidence-based practice among health care professionals in Ethiopia: a systematic review and meta-analysisPLOS ONE

Dear Dr. Zewdie,

Thank you for submitting your manuscript to PLOS ONE. After careful consideration, we feel that it has merit but does not fully meet PLOS ONE’s publication criteria as it currently stands. Therefore, we invite you to submit a revised version of the manuscript that addresses the points raised during the review process.

We look forward to receiving your revised manuscript.

Kind regards,

Mulu Tiruneh

Academic Editor

PLOS ONE

Journal Requirements:

Reviewers' comments:

Reviewer's Responses to Questions

**Comments to the Author**

1. If the authors have adequately addressed your comments raised in a previous round of review and you feel that this manuscript is now acceptable for publication, you may indicate that here to bypass the “Comments to the Author” section, enter your conflict of interest statement in the “Confidential to Editor” section, and submit your "Accept" recommendation.

Reviewer #2: All comments have been addressed

Reviewer #3: (No Response)

2. Is the manuscript technically sound, and do the data support the conclusions?

Reviewer #2: Yes

Reviewer #3: (No Response)

3. Has the statistical analysis been performed appropriately and rigorously? 

Reviewer #2: Yes

Reviewer #3: (No Response)

4. Have the authors made all data underlying the findings in their manuscript fully available?

Reviewer #2: Yes

Reviewer #3: (No Response)

5. Is the manuscript presented in an intelligible fashion and written in standard English?

Reviewer #2: Yes

Reviewer #3: (No Response)

6. Review Comments to the Author

Reviewer #2: (No Response)

Reviewer #3: Table 1: Show more characteristics of studies, such as age, sex and etc.

It should be “sample size” instead of “sample”.

How do you define quality of study as “good”?

Table 2: Different studies may include different covariates in logistic regression. Simply pooling the adjusted odds ratios coming from different regression models may cause bias. List the details of the regression from each study to evaluate whether those models are comparable.

Page 6: better say “The secondary outcomes were factors associated with the evidence based

practice of health professionals.”

Figure 5 is not necessary and can be omitted.

Flowchart: list each reason and N of exclusion. It is not clear to state “Exclude by title or by reading abstract”.

Conclusion: when you say the prevalence of EBP was low, are you comparing it to other African countries or western countries?

7. PLOS authors have the option to publish the peer review history of their article (what does this mean?). If published, this will include your full peer review and any attached files.

Reviewer #2: **Yes: **Fentaw Teshome Dagnaw

Reviewer #3: No

---

## [Author Response · Author response to Decision Letter 2]

28 Aug 2023

A point-by-point response to reviewers and editor

First of all, we would like to say thank you to both the reviewers and editor, we are so lucky to have you who had a great contribution to the improvement of this work. Saying this we have addressed the given comments as follows.

Journal requirements 

Response: Ok thank you for the suggestion. This suggestion was given previously and we have addressed it and submitted our response and revised version. As we have suggested we have reviewed our reference and corrected. As our study is systematic review and meta-analysis to minimize publication bias we have included primary studies which are not published. So it seems retracted articles but not. To solve this we have indicated the URL of the web page in which the studies were available. We have put the URL for reference 27, 28, 30, 41, 43 as they were not published but available at research repositories and other webpages and indicate reference 38 as it was a thesis. Additionally reference 20 and 45 were not correctly cited in the previous manuscript version and we have correctly cited those references. 

Reviewer # 3 

Comment: Table 1: Show more characteristics of studies, such as age, sex and etc.

Response: Ok thank you for your comment. We have tried to extract other characteristics of study participants of included primary studies but some studies missed and others may report and also the categories might not be the same across the included primary studies. For example the age category of study participants in the primary studies is not uniform so to present it in table form as Table 1not feasible. So we have tried to include important characteristics of primary studies that are linked with the study outcome and common for all included studies.

Comment: It should be “sample size” instead of “sample”.

Response: Ok thank you for your comment. We accept the reviewer comment and correct it as the reviewer suggested as shown in the track change in the revised manuscript. 

Comment: How do you define quality of study as “good”? 

Response: Ok thank you for the issue raised. The quality of included studies is evaluated using the Newcastle Ottawa Quality Assessment scale score. According to the scale grading if the sum of scale score is greater than or equal to 7 the study is classified as good and if below 7 it is graded as poor. In our case, the scale score of all included primary studies lies from 8 to 9 which is good. However we have not explained how we graded the quality of included studies in the introductory paragraph that is our mistake. We accept the review comment and explain how we grade the quality of included studies in the revised manuscript as shown in track change. 

Comment: Table 2: Different studies may include different covariates in logistic regression. Simply pooling the adjusted odds ratios coming from different regression models may cause bias. List the details of the regression from each study to evaluate whether those models are comparable.

Response: Ok thank you for the issue raised. In identification of determinants of EBP utilization we have used factors which are associated in two or more primary studies which report adjusted odd ratio (AOR) which are output of multivariable logistic regression. All adjusted odd ratio (AOR) of primary studies which we have used in identification the overall determinants of EBP are output of multivariable logistic regression. However we have not describe which regression model output we have used and the reviewer concern is right and we accept the reviewer suggestion and make it clear in the revised manuscript as shown in the track change.

Comment: Page 6: better say “The secondary outcomes were factors associated with the evidence based practice of health professionals.”

Response: ok thank you for your comment. We accept your comment and rewrite it as shown in the track change of the revised manuscript.

Comment: Figure 5 is not necessary and can be omitted.

Response: ok thank you for your comment. However Figure 5 is about sensitivity analysis result of our review which shows whether single study dominated the overall pooled prevalence of EBP utilization of health professional or not. Unless it is graphically presented and explained there is no statistical test which prove the sensitivity analysis of our review; so we have decided to present figure 5 as graphical explanation of sensitivity analysis. 

Comment: Flowchart: list each reason and N of exclusion. It is not clear to state “Exclude by title or by reading abstract”

Response: ok thank you for your comment. We have used the updated Preferred Reporting Items for Systematic Review and Meta-Analysis flow chart. In the flow chart exclusion by title and abstract has no reason because it is made before assessment by eligibility criteria. The reason for exclusion by title and by abstract is made if the title and abstract of the article is not complimentary with the current topic under review it would be excluded. Reason for exclusion is listed in the step of screening after full text articles are evaluated against the eligibility criteria then the number of article excluded and the respective reason for exclusion is explained. So we have explained the reason for the exclusion of 25 articles as the outcome of interest is not reported. 

Comment: Conclusion: when you say the prevalence of EBP was low, are you comparing it to other African countries or western countries?

Response: ok thank you for the issue raised. As we know the conclusion of a scientific paper is written after we compare and contrast our finding with the existing finding by others. Conclusion is judgmental summarization of the current finding with in light of the existed evidence and expected standard. So in our review what we have done is we summarizes our finding / the prevalence of good EBP/ by comparing it with other finding that we made in discussion section and with the expected level of EBP utilization. Since conclusion summarize all contents of a scientific paper we are aiming to include the points in the discussion too. We understand the reviewer concern and try to make the conclusion more informative and clear as shown in track change in the revised manuscript. Once again thank you very much.

---

## [Decision Letter · Decision Letter 3]

16 Oct 2023

PONE-D-23-09153R3Determinants of evidence-based practice among health care professionals in Ethiopia: a systematic review and meta-analysisPLOS ONE

Dear Dr. Zewdie,

Thank you for submitting your manuscript to PLOS ONE. After careful consideration, we feel that it has merit but does not fully meet PLOS ONE’s publication criteria as it currently stands. Therefore, we invite you to submit a revised version of the manuscript that addresses the points raised during the review process.

ACADEMIC EDITOR:

Comments

Topics-needs modification (which health care profession? Nurse, medical professional, midwifery). No need mixing these health professionals in oneQuality assessment -Add quality assessment of the included article as supplement filesHow to calculated pooled AOR using AOR of each study?Please ensure that your decision is justified on PLOS ONE’s publication criteria and not, for example, on novelty or perceived impact.

We look forward to receiving your revised manuscript.

Kind regards,

Atalel Fentahun Awedew, MD,MPH

Academic Editor

PLOS ONE

Journal Requirements:

Reviewers' comments:

Reviewer's Responses to Questions

**Comments to the Author**

1. If the authors have adequately addressed your comments raised in a previous round of review and you feel that this manuscript is now acceptable for publication, you may indicate that here to bypass the “Comments to the Author” section, enter your conflict of interest statement in the “Confidential to Editor” section, and submit your "Accept" recommendation.

Reviewer #3: (No Response)

2. Is the manuscript technically sound, and do the data support the conclusions?

Reviewer #3: (No Response)

3. Has the statistical analysis been performed appropriately and rigorously? 

Reviewer #3: (No Response)

4. Have the authors made all data underlying the findings in their manuscript fully available?

Reviewer #3: (No Response)

5. Is the manuscript presented in an intelligible fashion and written in standard English?

Reviewer #3: (No Response)

6. Review Comments to the Author

Reviewer #3: I thought the results of Figure 5 can be explained in the text. If the authors insist to add figure 5, please add more details about what the figure is presenting and what the interpretation is for general audience.

7. PLOS authors have the option to publish the peer review history of their article (what does this mean?). If published, this will include your full peer review and any attached files.

Reviewer #3: No

---

## [Author Response · Author response to Decision Letter 3]

19 Oct 2023

A point-by-point response to reviewers and editor

First of all, we would like to say thank you to both the reviewers and editor, we are so lucky to have you who had a great contribution to the improvement of this work. Saying this we have addressed the given comments as follows.

Journal requirements 

Response: Ok thank you for the suggestion. This suggestion was given previously and we have addressed it and submitted our response in the revised version. As we have suggested we have reviewed our reference and corrected. As our study is systematic review and meta-analysis to minimize publication bias we have included primary studies which are not published. So it seems retracted articles but not. To solve this we have indicated the URL of the web page in which the studies were available. 

ACADEMIC EDITOR Comments

Comment: Topics-needs modification (which health care profession? Nurse, medical professional, midwifery). No need mixing these health professionals in one

Response: Ok thank you for your comment. Our article is a systematic review and meta-analysis which reviews primary studies and synthesize evidence. So those primary studies included in our review were conducted among health care professionals including nurses, medical professionals, midwifes and other health professionals together as study participant in each primary studies. That means one primary study include nurses, medical doctors, midwifes and other health professionals together as study participant. So it was limitation of those included primary studies. While we do the meta-analysis we are unable to specify to specific professionals since we cannot get the level of evidence based practice of those specific professionals because those the primary studies has gross level of evidence based practice of the health professional as a whole. Even if it was limitation of primary studies, we understand the editor concern and try to put it as limitation of our review as shown in highlighted text in the revised manuscript.

Comment: Quality assessment -Add quality assessment of the included article as supplement files

Response: Ok thank you for your comment. We have added the modified Newcastle Ottawa Quality Assessment Scale for cross-sectional studies as supplementary file labeled as S2 Table and also cited in the manuscript in line 143. 

Comment: How to calculated pooled AOR using AOR of each study?

Response: Ok thank you for your question. Pooled AOR can be calculated using two methods. The first is using AOR of primary studies; second using number of events and number of subjects in each group. In our review we have used the first method sometimes known as the conventional method. In this method we extract the point estimate of AOR, the upper and lower confidence intervals of AOR of each studies for respective variable. Then the logarithm of those extracted indices is generated using STATA. Then the logarithm of those extracted indices used as effect size in each study, along with an estimate of its variance. Those estimated variances are usually assumed to be equal to the true variances. Finally using the metan command we have generated the pooled adjusted odd ratio on STATA 17. 

Reviewer # 3 

Comment: Reviewer #3: I thought the results of Figure 5 can be explained in the text. If the authors insist to add figure 5, please add more details about what the figure is presenting and what the interpretation is for general audience.

Response: Ok thank you for your comment. We accept the reviewer suggestion and omit figure 5 and explain the result of the figure by the text in the revised version of the manuscript. 

Once again we want to say thank you very much.

---

## [Editor Report · Decision Letter 4]

23 Oct 2023

Determinants of evidence-based practice among health care professionals in Ethiopia: a systematic review and meta-analysis

PONE-D-23-09153R4

Dear Mr.Amare Zewdie

We’re pleased to inform you that your manuscript has been judged scientifically suitable for publication and will be formally accepted for publication once it meets all outstanding technical requirements.

Kind regards,

Atalel Fentahun Awedew, MD,MPH

Academic Editor

PLOS ONE

---

## [Editor Report · Acceptance letter]

31 Oct 2023

PONE-D-23-09153R4 

Determinants of evidence-based practice among health care professionals in Ethiopia: a systematic review and meta-analysis 

Dear Dr. Zewdie:

I'm pleased to inform you that your manuscript has been deemed suitable for publication in PLOS ONE. Congratulations! Your manuscript is now with our production department. 

Kind regards, 

on behalf of

Dr. Atalel Fentahun Awedew 

Academic Editor

PLOS ONE